

# Validation of automated paper screening for esophagectomy systematic review using large language models

Rashi Ramchandani[1,2], Eddie Guo[3], Esra Rakab[1], Jharna Rathod[1], Jamie Strain[4], William Klement[4,5], Risa Shorr[6], Erin Williams[7,8], Daniel Jones[7,8] and Sebastien Gilbert[7,8]

[1] Department of Medicine, University of Ottawa, Ottawa, Ontario, Canada
[2] Institute of Health Policy, Management and Evaluation, University of Toronto, Toronto, Ontario, Canada
[3] Cumming School of Medicine, University of Calgary, Calgary, Alberta, Canada
[4] Ottawa Hospital Research Institute, Ottawa, Ontario, Canada
[5] Faculty of Computer Sciences, Dalhousie University, Dalhousie, Halifax, Canada
[6] Library and Learning Services, The Ottawa Hospital, Ottawa, Ontario, Canada
[7] Division of General Surgery, Department of Surgery, The Ottawa Hospital, Ottawa, Ontario, Canada
[8] Division of Thoracic Surgery, Department of Surgery, The Ottawa Hospital, Ottawa, Ontario, Canada

Corresponding author
Rashi Ramchandani,
rashi.ramchandani@uottawa.ca

## ABSTRACT

**Background:** Large language models (LLMs) offer a potential solution to the labor-intensive nature of systematic reviews. This study evaluated the ability of the GPT model to identify articles that discuss perioperative risk factors for esophagectomy complications. To test the performance of the model, we tested GPT-4 on narrower inclusion criterion and by assessing its ability to discriminate relevant articles that solely identified preoperative risk factors for esophagectomy.
**Methods:** A literature search was run by a trained librarian to identify studies ($n = 1,967$) discussing risk factors to esophagectomy complications. The articles underwent title and abstract screening by three independent human reviewers and GPT-4. The Python script used for the analysis made Application Programming Interface (API) calls to GPT-4 with screening criteria in natural language. GPT-4's inclusion and exclusion decision were compared to those decided human reviewers.
**Results:** The agreement between the GPT model and human decision was 85.58% for perioperative factors and 78.75% for preoperative factors. The AUC value was 0.87 and 0.75 for the perioperative and preoperative risk factors query, respectively. In the evaluation of perioperative risk factors, the GPT model demonstrated a high recall for included studies at 89%, a positive predictive value of 74%, and a negative predictive value of 84%, with a low false positive rate of 6% and a macro-F1 score of 0.81. For preoperative risk factors, the model showed a recall of 67% for included studies, a positive predictive value of 65%, and a negative predictive value of 85%, with a false positive rate of 15% and a macro-F1 score of 0.66. The interobserver reliability was substantial, with a kappa score of 0.69 for perioperative factors and 0.61 for preoperative factors. Despite lower accuracy under more stringent criteria, the GPT model proved valuable in streamlining the systematic review workflow. Preliminary evaluation of inclusion and exclusion justification provided by the GPT model were reported to have been useful by study screeners, especially in resolving discrepancies during title and abstract screening.

**Conclusion:** This study demonstrates promising use of LLMs to streamline the workflow of systematic reviews. The integration of LLMs in systematic reviews could lead to significant time and cost savings, however caution must be taken for reviews involving stringent a narrower and exclusion criterion. Future research is needed and should explore integrating LLMs in other steps of the systematic review, such as full text screening or data extraction, and compare different LLMs for their effectiveness in various types of systematic reviews.

## HIGHLIGHT BOX

**Key findings**

- The GPT-4 model's decision (GPT model) had an 85.58% agreement rate with human reviewers' decision for the for perioperative and 78.75% for preoperative factors iteration.
- For the perioperative risk factors search, the GPT model had an AUC of 0.87, a recall score of included articles of 89% and a false positive rate of 6%. For identifying preoperative risk factors the GPT model had an AUC of 0.75, a slightly lower recall score of included articles of 67% and a false positive rate of 15%.
- The GPT model achieved a kappa-score of 0.69 for identifying perioperative risk factors, demonstrating substantial agreement with human reviewers' decision but moderate agreement for identifying preoperative risk factors.

**What is known and what is new?**

- It was previously known that the advent of new large language models (LLMs) can streamline workflow, reduce costs, and improve efficiency in various areas of healthcare.
- The results of this study demonstrate promising use for the GPT model in streamlining workflow and improving efficiency for title and abstract screening in a systematic review.

**What is the implication, and what should change now?**

- This study validates the potential use of LLMs to improve efficiency of systematic review workflow without compromising quality.
- Future research should explore the performance of LLMs and their applications in other steps of the systematic review process, such as full-text screening or data extraction.
- There is a need for continuous refinement of LLMs and cautioned use given their constantly improving performance.

## INTRODUCTION

Systematic reviews are essential in synthesizing evidence to address specific clinical questions, potentially generating practice changing information in the process. Conducting a systematic review is a meticulous process involving several steps, each of which must be done methodically to avoid introduction of biases that may mislead conclusions and the overall validity of the study (*Harris et al., 2014*). To help with this, guidelines, such as the COCHRANE handbook, and more specifically the Preferred Reporting Items for Systematic Reviews and Meta-Analyses (PRISMA) framework for systematic reviews, have been created to ensure methodological consistency (*Page et al., 2021*). This process typically includes a thorough database search, followed by manual title and abstract screening, full-text assessment, data extraction, and risk of bias evaluation. Each step is performed by at least two independent reviewers to enhance reliability. At each step interrater reliability is compared, discrepancies are discussed and resolved with discussion or consultation with a third reviewer who is usually an expert in the field (*Khan et al., 2003*; *Tawfik et al., 2019*). Despite this process, challenges persist, including the time and labor intensiveness of each step. These hurdles can greatly prolong the review timeline, sometimes spanning months to years (*Clark et al., 2021*).

One significant challenge is ensuring sufficient interrater reliability, particularly in resolving discrepancies among reviewers, which can further extend the review duration. To mitigate these challenges and expedite the process, large language models (LLM) are increasingly being integrated to automate steps of the systematic review process. In fact, artificial intelligence-based models such as Research Screener, Abstrackr, and DistillerSR have shown promise in automating certain review tasks, substantially reducing workload and time expenditure (*Clark et al., 2021*; *Datt et al., 2024*; *Briganti, 2024*). For instance, Research Screener has demonstrated workload savings of up to 96% across various systematic reviews. However, despite their ability to streamline steps of a typically lengthy systematic review process, these models still pose computational and logistical challenges due to their reliance on semi-automated rather than fully automated tasks. Specifically, users need to be knowledgeable about the algorithms driving these models, and they need to learn to appropriately input abstracts for screening and correctly interpret the output results.

There are also inherent risks associated with these models, such as the potential omission of relevant studies. This risk is particularly pronounced when using n-gram-based approaches, which analyze fixed-length sequences of tokens using count-based probabilities (*Datt et al., 2024*; *Briganti, 2024*). In contrast, the generative pre-trained transformers (GPT) is a model that allows for prediction of a next token from previous tokens, enabling it to capture long-range dependencies and contextual relationships with natural language, thereby providing more context-aware analyses compared to traditional n-gram approaches (*Datt et al., 2024*; *Briganti, 2024*). While there are existing limitations in LLMs, namely the difficulty of transfer learning, advancements have been made in newer AI models, such as OpenAI's GPT-4 model, which offers enhanced capabilities in natural language processing and understanding. In the context of

systematic reviews, LLMs offer the potential to streamline labor-intensive tasks and reduce the workflow for investigators (*Chai et al., 2021*; *Hamel et al., 2020*; *Polanin et al., 2019*; *Nussbaumer-Streit et al., 2021*). In fact, recently, *Guo et al. (2024)* proposed leveraging the OpenAI for systematic review screening and demonstrated promising results, achieving a macro F1 score of 0.60 and a sensitivity of 0.76 for included articles. Despite the availability of more advanced models, broad acceptance and convenience of LLMs for users were key factors in its selection for examining systematic review processes in this study.

Therefore, in this study we sought to assess the ability of the GPT-4 model in discriminating relevant and non-relevant articles, as measured by the area under the curve (AUC), for title and abstract screening for a thoracic surgery-related systematic review focused on uncovering perioperative risk factors for esophagectomy complications. Perioperative factors encompass aspects occurring before (pre), during (intra), and after (post) the surgical procedure that can impact esophagectomy outcomes. To rigorously evaluate this method, in a subsequent analysis we applied a more stringent (*i.e.*, narrow) inclusion criterion, specifically focusing only on preoperative factors related to esophagectomy complications. Unlike other studies examining the use of LLMs in the systematic review process, our study uniquely assesses the model's performance across two distinct inclusion criteria—one broad (perioperative risk factors) and one narrow (preoperative risk factors). This approach enabled us to rigorously evaluate the effectiveness of the GPT-4 model under varied levels of stringency, providing insights into its adaptability and robustness in more challenging conditions. As such, this article sought to answer the research question of: How effective is the OpenAI GPT-4 model in identifying relevant studies during the title and abstract screening stage of systematic reviews, particularly when comparing broad (perioperative) *vs* narrow (preoperative) inclusion criteria for perioperative risk factors in esophagectomy?

## METHODS

The methodology for our systematic review included a comprehensive search of EMBASE, Medline, and the Cochrane Library from inception to February 21, 2023. The search strategy combines Medical Subject Headings (MeSH) terms and keywords related to esophagectomy, complications following esophagectomy, and risk factors for complications. The search strategy was designed and validated by a trained research librarian (RS) to identify studies that investigate perioperative risk factors for complications following esophagectomy (Appendix 1). The final step involved combining the search results related to esophagectomy and postoperative complications with those related to risk factors to identify relevant studies for inclusion in the systematic review.

From the database search a total of 2,242 articles were included. After duplicates were removed a total of 1,763 articles entered the title and abstract screening stage. All title and abstracts were manually screened by two independent reviewers (R.R or J.R, E.R) using Covidence, an online software for systematic reviews. Studies were excluded if they involved animal models, were not written in English, did not include mentioned outcomes, or were reviews, position statements or conference abstracts. Throughout the screening,

screeners solely selected the final decision of inclusion or exclusion, without providing any qualitative justification. Discrepancies among human reviewers were resolved with discussion with consultation with a field expert when necessary.

The Python script created by *Guo et al. (2024)* was adopted to make calls to the OpenAI application programming interface (API) with the prespecified screening inclusion and exclusion criteria in natural language alongside the literature search results that were manually screened by the human reviewers. The screening inclusion and exclusion criteria (Appendix 2) was fed to the Python script which would subsequently be fed to the GPT-4 API. An example of the script input is provided in Table 1. The GPT model was employed without additional fine-tuning, using default parameters of temperature = 0.7, max tokens = 2,048, top p = 1.0, frequency penalty = 0, and presence penalty = 0. The GPT-4 API was used to analyze and generate text based on the screening inclusion and exclusion criteria provided to it *via* prompting. A consistent instruction prompt was passed to GPT-4 for each article to determine suitability for inclusion based on the prespecified inclusion and exclusion criteria (*Polanin et al., 2019*). For this investigation, title and abstract screening was conducted by inputting the literature search results into a Python script, created by *Guo et al. (2024)*, which made automated calls exclusively to the GPT-4 API. At no point was the ChatGPT interface directly used for screening decisions. The Python script was queried twice on the literature search results. During the first query, the inclusion criteria encompassed studies discussing perioperative risk factors influencing esophagectomy complications. In the second query, the inclusion criteria were narrowed to focus on pre-operative risk factors (Fig. 1). The two runs of the model—one using broader perioperative inclusion criteria and the other with a narrower preoperative inclusion criterion— were conducted to assess how a more stringent inclusion standard might impact the LLM's performance and its ability to accurately identify relevant studies under varied levels of specificity.

Multi-level analysis was conducted to evaluate the GPT model's ability to correctly identify relevant articles and exclude irrelevant ones. Initially, the agreement percentage between the GPT model and human reviewers was determined. Subsequently, precision was calculated as the ratio of articles correctly identified as relevant (true positives) by the GPT model to all the articles identified as relevant (true positives and false positives). Recall was then calculated by dividing the articles correctly identified as relevant by the GPT model to all the articles that were actually relevant according to human reviewers (true positives and false negatives). The precision and recall values were used to calculate the macro F1 score, providing an overall assessment of the model's performance across the binary classes of relevant and non-relevant articles. Additionally, the area under the curve (AUC) was computed using a threshold of 0.5, to provide insight on the GPT model's ability to distinguish between relevant and non-relevant articles. False positive rate for both included and excluded articles, along with interrater reliability using Cohen's κ and prevalence-adjusted and bias-adjusted κ (PABAK), were also calculated against the human-reviewed articles. In the analysis, the final inclusion or exclusion decision reached by the three human reviewers was regarded as "the ground truth" or as the true positive and true negative values.

**Table 1  Example of input and output for GPT model depicting input and output decision.**

| GPT-model input | | | | Human decisions | | GPT-model output | |
|---|---|---|---|---|---|---|---|
| Authors | Year | Title | Abstract | Human reviewer 1 decision | Human reviewer 2 decision | AI decision | AI reasoning |
| Zhang et al. (2022) | 2023 | Research Progress on the Predicting Factors and Coping Strategies for Postoperative Recurrence of Esophageal Cancer | Esophageal cancer is one of the malignant tumors with poor prognosis in China. Currently, the treatment of esophageal cancer is still based on surgery, especially in early and mid-stage patients, to achieve the goal of radical cure. However, esophageal cancer is a kind of tumor with a high risk of recurrence and metastasis, and locoregional recurrence and distant metastasis are the leading causes of death after surgery. Although multimodal comprehensive treatment has advanced in recent years, the prediction, prevention and treatment of postoperative recurrence and metastasis of esophageal cancer are still unsatisfactory. How to reduce recurrence and metastasis in patients after surgery remains an urgent problem to be solved. Given the clinical demand for early detection of postoperative recurrence of esophageal cancer, clinical and basic research aiming to meet this demand has been a hot topic, and progress has been observed in recent years. Therefore, this article reviews the research progress on the factors that influence and predict postoperative recurrence of esophageal cancer, hoping to provide new research directions and treatment strategies for clinical practice. Copyright −© 2022 by the authors. | Exclude | Exclude | Exclude | Reasoning: the abstract indicates that the article is a review of research progress on factors influencing and predicting postoperative recurrence of esophageal cancer. it does not specifically mention perioperative risk factors associated with anastomotic complications or other complications such as aspiration pneumonia, delayed gastric emptying, malnutrition, dilation, Mendelson syndrome, and prolonged length of stay or hospitalization following esophagectomy. instead, it focuses on recurrence and metastasis after surgery. additionally, the article is a review article, which is explicitly excluded by criterion 3 of the exclusion criteria. |
| Yoshida et al. (2023) | 2023 | Omental Flap Wrapping Around the Esophagogastric Anastomosis and Association with Anastomotic Leak in Esophagectomy for Esophageal Cancer: A Propensity Score-Matching Analysis | BACKGROUND: Anastomotic leakage after esophagectomy remains frequent and can cause surgery-related mortality. Wrapping the esophageal conduit anastomosis with a greater omental flap is a proposed method to reduce leakage. However, the usefulness of omental wrapping (OW) has not been elucidated. STUDY DESIGN: This study included 338 patients with esophageal cancer who underwent McKeown esophagectomy and reconstruction using a gastric conduit between April 2005 and August 2021. The study participants were divided into 2 groups: reconstructions with and without OW. We performed propensity score matching between the reconstructions with and without OW. The propensity score was calculated according to baseline characteristics, performance and physical statuses, presence of comorbidities, and types and details of the treatment procedures undergone., RESULTS: We included 338 patients (169 with OW and 169 without OW) in this study. After matching, all clinical and surgical features were statistically equivalent between the groups. Reconstruction and anastomosis with OW were significantly associated with decreased leakage ($p = 0.016$) and surgical site infection ($p = 0.041$). Reconstruction and anastomosis with OW also exhibited a trend toward more frequent stricture, although this was not statistically significant ($p = 0.051$). Independent risk factors for leakage were reconstruction without OW (odds ratio, 2.55; 95% CI [1.234–5.285]; $p = 0.0088$) and younger age (odds ratio, 1.06; 95% CI [1.012–1.102]; $p = 0.011$), CONCLUSIONS: OW can reduce leakage in the esophagogastric anastomosis. OW can be performed conveniently and may be recommended to reinforce the anastomotic site, which can improve short-term outcomes after esophagectomy for esophageal cancer. Copyright −© 2022 by the American College of Surgeons. Published by Wolters Kluwer Health, Inc. All rights reserved. | Include | Include | Include | Reasoning: the articles title and abstract indicate that it is a study focusing on patients who underwent esophagectomy for esophageal cancer, which meets criterion 1. it reports on perioperative risk factors associated with anastomotic complications, specifically anastomotic leaks and potential strictures following esophagectomy, aligning with criterion 2. although the abstract does not explicitly mention other complications such as aspiration pneumonia, delayed gastric emptying, malnutrition, dilation, mendelson syndrome, or prolonged length of stay or hospitalization, the focus on anastomotic leaks, which are a significant complication, suggests relevance to criterion 3. the study design is a propensity score-matching analysis, which is a type of observational study, thus fulfilling criterion 4. the article is published in English as per criterion 5 and involves adult participants as indicated by the context of esophageal cancer surgery, satisfying criterion 6. The study does not fall under any of the exclusion criteria: it reports specific perioperative risk factors (exclusion criterion 1), it does not appear to be a case report, review, letter, editorial, or conference abstract (exclusion criterion 3), it is not focused solely on a pediatric population (exclusion criterion 4), there is no indication of insufficient data or incomplete reporting (exclusion criterion 5), it is relevant to the investigation of anastomotic complications after esophagectomy (exclusion criterion 6), there is no specified date before which studies are excluded (exclusion criterion 7), it is not a non-human study or conducted on cadavers (exclusion criterion 8), and there is no indication that it is a duplicate publication or multiple reports from the same study (exclusion criterion 9). |

| GPT-model input | | | | Human decisions | | GPT-model output | |
|---|---|---|---|---|---|---|---|
| Authors | Year | Title | Abstract | Human reviewer 1 decision | Human reviewer 2 decision | AI decision | AI reasoning |
| Yang et al. (2023) | 2023 | Comparison of neoadjuvant nab-paclitaxel plus immunotherapy vs paclitaxel plus immunotherapy for esophageal squamous cell carcinoma | BACKGROUND: This study aimed to compare the feasibility of nab-paclitaxel plus platinum-based chemotherapy (nabTP) vs paclitaxel plus platinum-based chemotherapy (TP) with immune checkpoint inhibitors (ICIs) as a neoadjuvant modality for locally resectable esophageal squamous cell carcinoma (ESCC). METHODS: Between April 2019 and March 2022, we identified ESCC patients who received neoadjuvant immunotherapy with both nabTP ($n = 213$) and TP ($n = 98$) at our institution and Henan Cancer Hospital. The patients in the ICIs-nabTP and ICIs-TP groups were pair-matched (1:1) for tumor location, sex, smoking, drinking, clinical T and N stage. The primary endpoint was the hazard of 30-day major postoperative complications. Second, logistic models were applied to estimate the risk factors for pathological complete response (pCR) rate. RESULTS: All patients underwent esophagectomy with R0 resection. A statistically significant increase in the risk of developing major pulmonary (odds ratio (OR), 1.182; 95% confidence interval (CI): [0.530–2.635]; $p = 0.683$), anastomotic (OR, 1.881; 95% CI [0.607–5.830]; $p = 0.267$), cardiac (OR, 1.000; 95% CI [0.426–2.349]; $p = 1.000$) complications after neoadjuvant immunotherapy plus nabTP was not observed. The median interval to surgery was 39 days in the ICIs-nabTP group vs 44 days in the ICIs-TP group ($p = 0.119$). There was no 30-day mortality in each group. However, there was a slight difference in the 30-day readmission rate ($p = 0.043$) and the incidence of hydropneumothorax ($p = 0.027$) between the two groups. The pCR rates of the ICIs-nabTP and ICIs-TP group were 36.7 and 21.4%, respectively ($p = 0.018$)., CONCLUSIONS: It appears to be feasible to add immunotherapy to nabTP regimen for locally advanced ESCC. Compared with TP, nabTP plus ICIs can achieve a better pCR rate in ESCC. Copyright –© 2023 The Authors. Thoracic Cancer published by China Lung Oncology Group and John Wiley & Sons Australia, Ltd. | Include | Exclude | Include | Reasoning: the study focuses on patients who underwent esophagectomy for esophageal squamous cell carcinoma, which meets criterion 1. it reports on perioperative risk factors associated with anastomotic complications, as well as other complications such as pulmonary and cardiac complications, and the incidence of hydropneumothorax, which aligns with criteria 2 and 3. the study design is not explicitly stated in the abstract, but it appears to be a cohort study given the comparison between two groups of patients, which would satisfy criterion 4. the article is published in English, meeting criterion 5, and involves adult participants, satisfying criterion 6. the study does not fall under any of the exclusion criteria: it reports specific perioperative risk factors (exclusion criterion 1), assesses risk factors for other complications (exclusion criterion 2), is not a case report, review, letter, editorial, or conference abstract (exclusion criterion 3), does not focus solely on a pediatric population (exclusion criterion 4), does not appear to have insufficient data or incomplete reporting based on the abstract (exclusion criterion 5), is relevant to the investigation of anastomotic complications after esophagectomy (exclusion criterion 6), and there is no specified date before which studies are to be excluded (exclusion criterion 7). it is not a non-human study or conducted on cadavers (exclusion criterion 8), and there is no indication that it is a duplicate publication or multiple reports from the same study (exclusion criterion 9). |

**Note:**
A csv file is loaded into the python script that makes API calls to the GPT-model and a csv file with the depicted columns is outputted after running the GPT-model. The GPT model was employed without additional fine-tuning, using default parameters of temperature = 0.7, max tokens = 2,048, top p = 1.0, frequency penalty = 0, and presence penalty = 0.

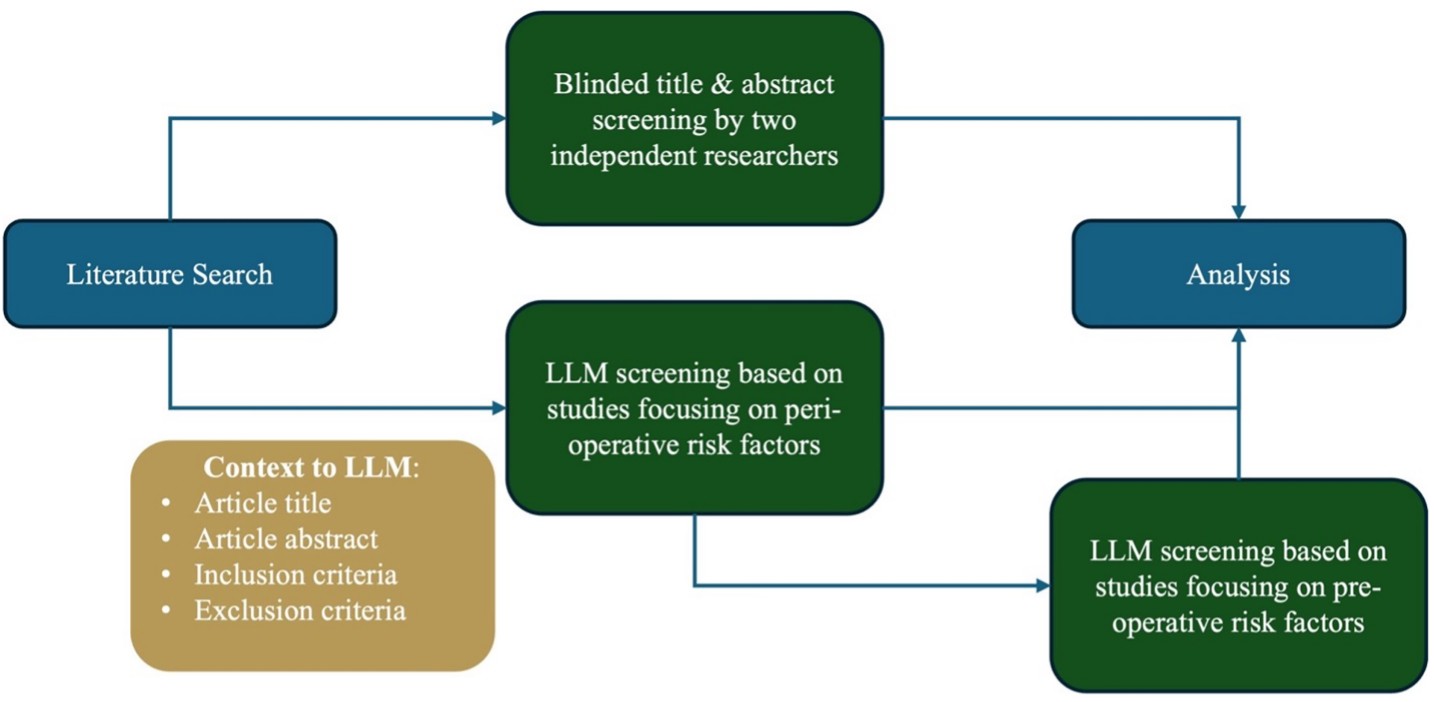

**Figure 1** **Visual depiction of study methodology from literature search to analysis.** Figure depicts the workflow and methodology of the study describing the two runs of the Python script with the broad (perioperative risk factors) and narrow (perioperative risk factors) screening.

## RESULTS

### Perioperative risk factor analysis

In this study, 1,967 studies were screened for title and abstract by two independent screeners as well as the GPT model. When screening with the perioperative inclusion criteria, the GPT model and human reviewers had 1,680 instances of agreement (both inclusion and exclusion decisions), resulting in an agreement rate of 85.58%, calculated as the ratio of agreements to the total number of studies (1,680/1,967). The GPT model demonstrated strong performance in identifying relevant articles, with a precision of 0.74 and a high recall of 0.89 (Table 2). The macro F1 score, which balances precision and recall, was 0.81 while the AUC value was 0.87 (Fig. 2). Furthermore, the GPT model demonstrated a negative predictive value (NPV) of 0.84 (Table 3). This suggests that when human reviewers decide to include a study, there is a 74% change that the GPT model would also include the article, whereas if the human reviewers decide to exclude an article, there is an 84% change the GPT model would exclude it as well. Additionally, the false positive rate for the GPT model was 0.06, indicating a low rate of incorrectly identifying non-relevant studies as relevant.

Moreover, the recall of excluded articles was 0.59, suggesting a moderate rate of correctly identifying non-relevant studies. The interobserver reliability score, known as the kappa statistic, for manual title and abstract screening by two independent human observers was 0.50, suggesting mild agreement.

**Table 2 Summary of key findings demonstrating model performance when evaluating perioperative and preoperative GPT model runs.**

| Metric | Formula | Value |
|---|---|---|
| Agreement | $\dfrac{\text{Number of agreements between GPT} - \text{model and human reviewers}}{\text{Total number of studies}}$ | Perioperative: 85.58%<br>Preoperative: 79.03% |
| Recall | $\dfrac{\text{True Positives}}{(\text{Total Positives} + \text{False Negatives})}$ | Perioperative: 0.89<br>Preoperative: 0.67 |
| Precision | $\dfrac{\text{True Positives}}{(\text{Total Positives} + \text{False Positives})}$ | Perioperative: 0.74<br>Preoperative: 0.65 |
| Macro F1 score | $2 \times \dfrac{(\text{Precision} \times \text{Recall})}{(\text{Precision} + \text{Recall})}$ | Perioperative: 0.81<br>Preoperative: 0.66 |
| Area under the curve (AUC) | Calcaulted using a threshold of 0.5 for distinguising relevant and non − relevant articles | Perioperative: 0.87<br>Preoperative: 0.75 |
| False positive rate | $\dfrac{\text{False Positives}}{(\text{False Positives} + \text{True Negatives})}$ | Perioperative: 0.06<br>Preoperative: 0.15 |
| Negative predictive value (NPV) | $\dfrac{\text{True Negatives}}{(\text{True Negatives} + \text{False Negatives})}$ | Perioperative: 0.84<br>Preoperative: 0.85 |
| Prevalence-adjusted bias-adjusted kappa (PABAK) | Adjusted Kappa to account for prevalence and bias in dataset | Perioperative: 0.70<br>Preoperative: 0.51 |
| Interobserver Kappa Statistic | $\dfrac{(\text{Observed Agreement} - \text{Expected Agreement})}{(1 - \text{Expected Agreement})}$<br>*Agreement between two independent human reviewers | Perioperative: 0.59<br>Preoperative: 0.57 |
| GPT K value | $\dfrac{(\text{Observed Agreement} - \text{Expected Agreement})}{(1 - \text{Expected Agreement})}$<br>*Agreement between final human reviewer decision and GPT decision | Perioperative: 0.69<br>Preoperative: 0.61 |
| Cost | Obtained data point | Perioperative: $29.43<br>Preoperative: $31.08 |
| Run time | Obtained data point | Perioperative: $21,130.50<br>Preoperative: $29,084.10 |

For the purposes of evaluating agreement, a "combined human decision" was defined as the consensus decision reached by the two independent human reviewers after resolving any discrepancies through discussion. This combined human decision was treated as one reviewer, and the GPT decision was treated as a second reviewer. Using this approach, the kappa statistic was calculated. The κ for the perioperative risk factors search was 0.69, indicating substantial agreement between the GPT model and the consensus human decision.

Furthermore, the PABAK for the perioperative analysis was 0.70, which provides a more accurate measure of agreement between the human raters' decision and the AI decision when dealing with an imbalanced data set.

The cost for the perioperative risk factor analysis was $29.43 USD, and the run time was 21,130.5 s.

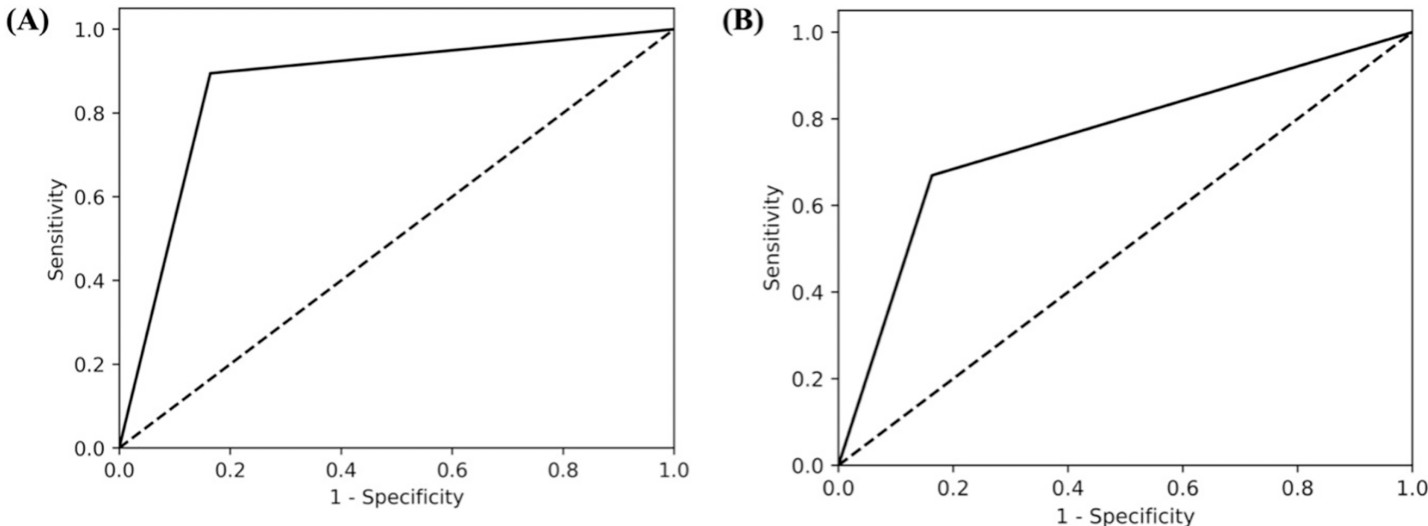

**Figure 2 AUC curves for (A) peri-operative and (B) pre-operative risk factors.** AUC score of perioperative run was 0.87, while for the pre-operative run it was 0.75 which indicates that the GPT-model is able to differentiate between relevant and non-relevant articles, particularly per-forming well above the threshold of random chance, which is typically considered to be 0.5.

**Table 3 2 × 2 table depicting true positives, false negatives, true negatives, and false positive for perioperative risk factors.**

|  |  | Human Decision | |
|---|---|---|---|
|  |  | Include | Exclude |
| GPT-model decision | Include | 30.36% (*n* = 596) | 10.85% (*n* = 213) |
|  | Exclude | 10.85% (*n* = 213) | 48.04% (*n* = 945) |

## Preoperative risk factor analysis

Similar to the perioperative iteration, the 1,967 search results underwent title and abstract screening by two independent reviewers and the GPT-model. However, this iteration a narrower inclusion criterion was applied for title and abstract screening such that solely articles that discussed preoperative risk factors (*i.e.*, factors before surgery) and their relation to esophagectomy complications were included.

From title and abstract screening, there were 1,549 agreements between the GPT model and human reviewers' decisions, yielding a total percentage accuracy of 78.75% (1,549/1,967). For preoperative risk factors, the GPT model demonstrated moderate performance in identifying relevant articles, with a precision of 0.65 and moderate recall of 0.67. The macro F1 score was 0.66 and the AUC was 0.75 (Fig. 2). Furthermore, the GPT model demonstrated a negative predictive value (NPV) of 0.85 (Table 4). This suggests that when human reviewers decide to include a study, there is a 65% change that the GPT model would also include the article, whereas if the human reviewers decide to exclude an article, there is an 85% change the GPT model would exclude it as well.

**Table 4 2 × 2 table depicting true positives, false negatives, true negatives, and false positive for preoperative risk factors.**

| | | Human decision | |
| --- | --- | --- | --- |
| | | Include | Exclude |
| GPT-model decision | Include | 20.51% (*n* = 402) | 11.23% (*n* = 221) |
| | Exclude | 10.92% (*n* = 197) | 58.52% (*n* = 1,147) |

The recall for excluded articles was 0.16. The interobserver kappa statistic for manual title and abstract screening by the human screeners was 0.57, indicating moderate agreement. Additionally, κ between human and GPT-4 decisions for the preoperative search was 0.61, suggesting substantial agreement. The Prevalence-Adjusted Bias-Adjusted Kappa (PABAK) for the preoperative analysis was 0.51. The false positive rate for the LLM was 0.15. The cost for the perioperative risk factor analysis was $31.08 USD, and the run time was 29,084.1 s.

In the broader run of the model, focused on identifying articles discussing perioperative risk factors, the LLM included 189 cases that it did not include in the narrower preoperative run, which targeted identifying articles that only mentioned preoperative risk factors. This suggests that these cases met the general perioperative criteria but not the more specific preoperative criteria. Conversely, the LLM included 70 cases in the narrower preoperative run that it excluded from the broader perioperative run, indicating these cases specifically addressed preoperative factors without fitting the broader perioperative criteria. Notably, manual screening by human reviewers did not show such discrepancies between the two sets of criteria reader.

### Length and qualitative assessment

Table 1 illustrates an example of the input and output for the GPT model. For the GPT model, a justification was requested for each inclusion or exclusion decision. The average justification length for the perioperative run was 1,449.23 characters, and for the preoperative run, it was 1,405.72 characters.

Qualitative inspection of the justifications provided by the GPT model was considered useful by the participating title and abstract screening reviewers. Preliminary discussions with the screeners found that justifications from the GPT model were insightful and generally thorough. Justifications of the GPT model often provided rationale for the inclusion or exclusion decision based on each of the criteria initially inputted into the model. However, at times, the rationale provided by the GPT model demonstrated significant shortcomings, such as misinterpreting or misaligning the criteria with its explanations. Examples included incorrect references to exclusion criteria or the introduction of nonexistent criteria, highlighting the model's limitations in accurately mapping its decisions to predefined rules. These errors emphasize the need for human oversight to verify the justifications provided by GPT.

## DISCUSSION

The findings of this study offer valuable insights into the reliability of LLMs in systematic review methodology, particularly when stringent inclusion and exclusion criteria are applied. To begin, the GPT model demonstrated a high level of agreement with human reviewers, with a percentage agreement of 85.58% for the perioperative run and 78.75% for the more specific preoperative run. The AUC values further support the model's performance, with both runs achieving AUC values above 0.5, being 0.87 for the perioperative run and compared 0.75 for the preoperative run. This indicates that the GPT model is able to differentiate between relevant and non-relevant articles, particularly performing well above the threshold of random chance, which is typically considered to be 0.5.

The analysis revealed more subtle distinctions in the model's performance when comparing its effectiveness with broader inclusion criteria of identifying perioperative factors, to its performance with a narrower inclusion criterion focused on identifying preoperative risk factors. For the perioperative run, the precision was 0.74 while the recall was 0.89. This indicates that the model identified 74% of the articles that were truly relevant. The corresponding macro F1 score of 0.81 suggests a good balance between precision and recall for the perioperative run. In contrast, the preoperative run showed slightly lower performance metrics. The precision for the preoperative run was also 0.65, while the recall was 0.67 indicating that the model only correctly identified 65% of relevant articles. This resulted in a lower macro F1 score, which is a harmonic mean of precision and recall, of 0.66 for the preoperative run compared to the perioperative run.

Insight of the model's performance can be drawn from the kappa score as well as it analyzes how much the model deviates from the "ground truth". There was substantial agreement when the GPT model attempted to identify perioperative risk factors ($\kappa = 0.69$), but only moderate agreement when identifying pre-operative risk factors ($\kappa = 0.57$). The reduced GPT-agreement with a narrower inclusion criterion is likely due to its lack of expertise to identify and interpret more specific surgical and medical terms. The reduced recall and agreement with narrower preoperative criteria likely result from GPT-4's difficulty interpreting clinically specific surgical terminology. This underscores the importance of further fine-tuning or training of the model with specialized medical terminology to enhance accuracy. Identifying preoperative risk factors often requires a more nuanced clinical understanding and specificity to recognize, interpret and recall relevant surgical terms. This discrepancy does not necessarily indicate a fundamental flaw in the machine learning algorithms themselves, but rather highlights the challenges the GPT model encounters with a lack of knowledge on surgical terms and limited clinical gestalt, which may not arise in manual screening by trained human reviewers who are experts in the field. Addressing this may require refining the training datasets or enhancing the model's ability to interpret and analyze more clinically detailed and specific data. In the meantime, caution should be advised for the use of LLMs for highly technical systematic reviews.

The recall of excluded articles was 0.59 for the perioperative run and 0.16 for the preoperative run, indicating limited promise in excluding irrelevant articles. Conversely, the recall of included articles varied depending on the run, with a recall rate of 0.89 for the perioperative run and 0.67 for the preoperative run. The high recall ability of the GPT model for the broader search of perioperative factors suggests that it is effective at identifying relevant studies and minimizing false negatives. This is supported by the low false positive rate of 0.06 for the perioperative run. However, the preoperative run showed a lower recall rating of included articles and a higher false positive rate of 0.15. Together, this suggests there is decreased accuracy for the GPT model when using more stringent criteria, indicating lower performance of the model under these specific conditions.

The evaluation of performance metrics in this study demonstrates the GPT models potential to complement human reviewers by effectively identifying relevant studies, especially in broader searches. While not flawless, the results highlight the promising potential of the GPT model to streamline the tedious process of title and abstract screening in a systematic review. The model's high agreement with human reviewers, coupled with its ability to distinguish between relevant and non-relevant articles, demonstrates its potential to streamline the systematic review process and reduce the burden of manual screening. However, there is poorer performance of the model for narrower inclusion/exclusion criteria and thus caution must be exercised.

## Potential of LLMs in streamlining systematic reviews

The integration of LLMs in systematic review tasks, such as title and abstract screening, has the potential to reduce human errors and bias (*Polanin et al., 2019*; *Nussbaumer-Streit et al., 2021*). In studies with a large number of articles, such as this investigation, human screeners can become fatigued and may not objectively evaluate studies. LLMs can help mitigate this issue, making the screening process easier for human reviewers. However, caution is warranted as human reviewers, especially experts in the field, can comprehend terminology and apply clinical gestalt when selecting studies. LLMs like GPT may lack this clinical gestalt component, which could impact decision-making regarding study inclusion/exclusion.

In the future, integrating LLMs can help prioritize workflow and reduce the number of human hours and associated costs for conducting systematic reviews, especially in cases with a large number of search results (*Nussbaumer-Streit et al., 2021*). By testing and eventually integrating LLMs into the review workflow, investigators and clinicians can focus on other tasks, such as data analysis, writing, and creating data extraction forms. While the exact time savings remain uninvestigated, this integration could significantly shorten the time needed to complete a systematic review and reduce burnout among reviewers.

The integration of LLMs in systematic reviews can also lead to cost savings. For example, in this study, the manual review of articles in this study took 50 h for one screener, totaling 100 h for both screeners. Considering a minimum wage of $16.55 CAD

in Ontario, the total compensation can be estimated to be approximately $1,655 CAD (16.55 * 100 h). This estimate does not account for additional costs such as employer contributions or benefits, which may further increase the hourly rate. For the GPT model, both runs for peri and preoperative risk factors were completed for a total of just 60.51 USD (82.79 CAD), resulting in savings in research funding. Applied on a broader scale, the money saved on human compensation could be redirected to fund more studies or expand the scope of existing studies, further enhancing the field.

## Comparison to existing models

*Guo et al. (2024)* compared the efficacy of the GPT model used in this study with other AI-based title and abstract screening models, including Research Screener, Abstrackr, and DistillerSR (*Polanin et al., 2019*). Compared to Abstrackr, which had an overall sensitivity of 0.91, this study demonstrated an inclusion sensitivity of 0.89 for the perioperative run, indicating comparable performance. However, relative to the GPT model, Abstrackr had a lower missed record percentage. In contrast, the GPT model had a lower missed record percentage compared to DistillerSR, which demonstrated up to a 100% missed study percentage in the last dataset (*Hamel et al., 2020*; *Hamel et al., 2021*).

Evidence to date also indicates significant variability in the performance of these models depending on the systematic review topic and the user's familiarity with the AI models (*Hamel et al., 2021*). The GPT-based approach used in this study simplifies the interface, as the Python script with an API call is publicly available to run on any device. The output is a Microsoft Excel file, which can be easily used by investigators or converted to CSV and integrated into systematic review software such as Covidence. While the computational demands of GPT are higher than Abstrackr or DistillerSR, due to its transformer architecture, it provides a user-friendly interface that simplifies setup and integration. Additionally, GPT's ability to generate explanatory justifications enhances interpretability, offering insights that can support reviewer decision-making.

While the LLM achieved accuracy rates of 85.58% and 79.03%, which indicate promising potential, these rates fall short of the 100% accuracy that might be ideal for independent decision-making in systematic reviews (*Hamel et al., 2021*; *Gates, Johnson & Hartling, 2018*). Given these limitations, LLMs are recommended as supportive tools in the systematic review process, particularly to aid in screening large datasets and resolving reviewer discrepancies rather than as standalone decision-makers. They are particularly useful in resolving discrepancies among reviewers by providing justifications that can support discussions and enhance consistency. Additionally, LLMs can streamline the initial screening process by filtering out irrelevant studies, reducing the workload for human reviewers. However, caution is advised, as these tools are not yet capable of fully replacing the accuracy and nuanced judgment of human reviewers in meticulous tasks such as title and abstract screening (*Tran et al., 2024*).

## Limitations and future studies

This study has some limitations regarding both internal and external validity. For internal validity, while the GPT model demonstrated a high agreement rate with human reviewers,

the accuracy of human screening decisions was used as the baseline, which may introduce subjectivity and affect the reliability of comparisons. Additionally, the model's performance may be influenced by specific inclusion criteria and the nature of the input data, which could limit the consistency of results across different systematic reviews. In terms of external validity, our findings are based on a systematic review focused on perioperative and preoperative risk factors for esophagectomy. As a result, the generalizability of the model's performance to other types of systematic reviews or medical topics may be limited.

An analysis of recurring words in the abstracts and titles of included studies revealed that terms such as "risk factors," "complications," "esophagectomy," and "perioperative" frequently appeared. However, these terms also appeared in many excluded studies, indicating that a simple keyword-based approach would likely result in a high false positive rate. This supports the use of a more sophisticated model like GPT-4, which leverages contextual understanding to differentiate relevant studies.

Another limitation with the study is the exclusion of articles that did not explicitly state outcomes. While this aligns with standard title and abstract screening practices, it could introduce bias in assessing the AI's performance relative to human reviewers. Human reviewers might still extract relevant insights from articles without stated outcomes, while the AI may rely more on explicit outcome statements to make inclusion or exclusion decisions. This could potentially affect the comparison between AI and human performance, as it may underestimate the AI's ability to interpret less straightforward abstracts.

Future studies can be conducted to examine whether a GPT-based approach, like the one used in this study, could be applied to other steps in the PRISMA guideline, such as for full-text screening or data extraction. Additional investigations can also be conducted to unveil a comprehensive comparison of the GPT model applied in this study with other existing AI models, such as Research Screener, Abstrackr, and DistillerSR (*Hamel et al., 2020*; *Hamel et al., 2021*). Such an investigation can include title and abstract screening for a variety of systematic reviews in the realm of clinical, basic, and translational sciences. Such an investigation could help determine whether the topic or area of focus of a review impacts the reliability of LLM-based screening. Another potential future area of research is domain-specific fine-tuning of LLMs to improve accuracy when applying stringent medical criteria, as well as exploring hybrid models that integrate human oversight to address limitations observed in this study.

## CONCLUSION

Overall, this study demonstrates the potential of LLMs, such as the GPT model employed in this investigation, to reduce the burden of review without introducing unnecessary results, particularly in title and abstract screening for thoracic surgery-related studies. Specifically, in response to our primary research question evaluating the effectiveness of GPT-4 in title and abstract screening for systematic reviews, we conclude that GPT-4 demonstrates high accuracy under broader inclusion criteria (perioperative factors; AUC = 0.87, agreement rate = 85.58%) but has reduced accuracy under narrower

inclusion criteria (preoperative risk factors), suggesting a cautious approach to its use in contexts requiring stringent specificity. These findings highlight its potential to complement human reviewers, particularly in broader systematic review tasks. Despite lower accuracy with stringent inclusion criteria, LLMs can serve as valuable tools for resolving discrepancies between human reviewers and for streamlining the systematic review workflow. Future research should aim to establish clearer guidelines on the appropriate use of LLMs based on inclusion criteria specificity and explore the integration of LLMs in other steps of the PRISMA guideline. Comparing the effectiveness of different LLMs across a variety of medical disciplines will further inform their best use cases in systematic reviews.

## ABBREVIATIONS

| | |
|---|---|
| **AI** | Artificial Intelligence |
| **GPT** | Generative Pre-trained Transformer |
| **LLM** | Large Language Model |
| **PRISMA** | Preferred Reporting Items for Systematic Reviews and Meta-Analyses |

### Funding
The authors received no funding for this work.

### Competing Interests
The authors declare that they have no competing interests.

### Author Contributions
- Rashi Ramchandani conceived and designed the experiments, performed the experiments, analyzed the data, prepared figures and/or tables, authored or reviewed drafts of the article, and approved the final draft.
- Eddie Guo conceived and designed the experiments, performed the experiments, analyzed the data, performed the computation work, prepared figures and/or tables, authored or reviewed drafts of the article, and approved the final draft.
- Esra Rakab performed the experiments, authored or reviewed drafts of the article, and approved the final draft.
- Jharna Rathod performed the experiments, authored or reviewed drafts of the article, and approved the final draft.
- Jamie Strain performed the experiments, authored or reviewed drafts of the article, assisted in research coordination, and approved the final draft.
- William Klement analyzed the data, prepared figures and/or tables, authored or reviewed drafts of the article, and approved the final draft.
- Risa Shorr conceived and designed the experiments, authored or reviewed drafts of the article, and approved the final draft.

- Erin Williams analyzed the data, authored or reviewed drafts of the article, and approved the final draft.
- Daniel Jones analyzed the data, authored or reviewed drafts of the article, and approved the final draft.
- Sebastien Gilbert analyzed the data, authored or reviewed drafts of the article, overall supervision/direction of project, and approved the final draft.

## Data Availability

The code to reproduce the work is available in the Supplemental File.

## Supplemental Information

Supplemental information for this article can be found online at http://dx.doi.org/10.7717/peerj-cs.2822#supplemental-information.

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
