# Peer review of "Validation of automated paper screening for esophagectomy systematic review using large language models"

_PeerJ Computer Science, doi:10.7717/peerj-cs.2822_

## Round 0.1 · original submission · Major Revisions

All 3 reviewers have requested Major Revisions - please respond to their comments in detail

Reviewer 1 ·

Basic reporting

Overall, the English is quite good, I only have a few minor suggestions.

1. Please streamline the lines 101-104 as they contain redundant information.
2. In line 106 you should introduce AUC as it is mentioned the first time
3. In line 133 it is "written in English"
4. I do not understand line 217-221. Can you make this more clear. What exactly happened?
5. Please provide evidence for line, 92-100, 137,and 141-144

Experimental design

The experimental design is interesting, although similar experiments have already been conducted multiple times across various disciplines including medicine.
See also: https://www.scopus.com/results/results.uri?sort=plf-f&src=s&st1=%22Large+language+models%22+AND+%22literature+review%22&sid=4a2bb102272246679387ac7ac6338073&sot=b&sdt=b&sl=62&s=TITLE-ABS-KEY%28%22Large+language+models%22+AND+%22literature+review%22%29&origin=searchbasic&editSaveSearch=&yearFrom=Before+1960&yearTo=Present&sessionSearchId=4a2bb102272246679387ac7ac6338073&limit=10

Please note the following:
6. Please formulate a specific research question and answer it.
7. Clearly identify the knowledge gap your study aims to address, especially in relation to other studies that have produced similar results.
8. Please provide more details what exactly constitutes a: True positive, false negative, false positive, true negative. Both, for testing human vs human and humans vs AI. In this regard, moving line 167-169 a bit up in the text could help.
9. Why does Table 2 sum up to n=2106 and Table 3 to 1960? I thought only 1967 papers in total were analyzed?

Validity of the findings

Overall, I consider the findings valid since they align with results from other studies. However, there are some aspects that remain unclear to me.

10. In line 172-175: You state 1967 studies were part of the study but you gain 1680 matches. How does this work? Also, how do you come up with the 85.58%?
11. Can you elaborate on line 186-190. What are the implications of that?
--> Two humans do not (mildly) agree with each other. But the combined human decision and AI do agree with each other quite well. However, when the human agreement is the baseline for determining accuracy, what does this tell us?
12. Explain how the combined human decision was formed. Especially in cases of disagreement.
13. In line 197-201: 1549 of 1967 is 78.74% not 79.03% Why the difference?
14. Please add a limitations section to your paper addressing issues with internal and external validity

Additional comments

I have some general points to make:

15. You excluded papers that did not state outcomes. However, excluding papers that did not state outcomes may significantly bias the results, as a GPT might perform much worse in those conditions compared to a human. A human could still derive useful insights from the paper by reading it. Please address this as a limitation.
16. 286-301 --> Is an accuracy of 85.58% or 79.03% enough for science? Shouldn't we strive for 100% What happened if an important part of information is lost in the 15% or 20%? Who is to blame? Can LLM really support science when scientist should opt for the correct result not a somewhat accurate result?
17. This research is not novel. However as stated in line 332-334 a full text analysis could significantly advance this topic. I strongly recommend pursuing that approach.

Reviewer 2 ·

Basic reporting

The manuscript is clearly written, and the language used is professional. However, there are several places where the clarity of the information could be improved. For example, the introduction section needs to be more specific when outlining the background of the study. More information about the current limitations of large language models (LLMs) in systematic reviews would provide better context. Additionally, the manuscript should clearly state why the GPT-model was chosen over other AI tools, especially in the context of esophagectomy. Figures and tables are well-labeled, but some need better captions to ensure clarity for readers unfamiliar with the topic.

Experimental design

While the methodology is generally well described, there are some areas where further clarification is needed. The study does not clearly justify why the inclusion and exclusion criteria were narrowed between the perioperative and preoperative factors. This gap makes it challenging to understand the rationale behind the different sensitivity levels. Additionally, the paper lacks sufficient details on how the GPT model was fine-tuned or adapted to this specific medical application. I am assuming everything is without fine-tuning. Providing more information on the parameters used for the GPT-model's screening process would increase the study's reproducibility.

Validity of the findings

The results appear promising, but the study does not fully explore the limitations of the GPT-model, particularly in handling complex medical terminology and nuances in the data. For example, the lower accuracy with preoperative factors suggests a potential issue with the model’s ability to discriminate between similar terms, which warrants more discussion. Moreover, the results are focused too heavily on numerical metrics (AUC, sensitivity, F1 scores), but there is limited discussion on how these findings translate into practical improvements for human reviewers. Adding more qualitative assessments from human screeners would strengthen the validity of the findings.

Additional comments

1. The conclusion should more clearly outline the implications of the findings, particularly regarding when LLMs should or should not be applied in systematic reviews involving highly specific medical criteria.
2. Future research directions should be elaborated, including more detailed suggestions on how to overcome the limitations of the current model (e.g., domain-specific fine-tuning).
3. The paper could benefit from a deeper comparison with existing models beyond the metrics, discussing trade-offs in terms of computational complexity, ease of use, and interpretability.

Reviewer 3 ·

Basic reporting

The paper discusses the potential of GPT to decide whether a paper meets the criteria to be included in a systematic review of the state of the art. The application is very useful, and the paper is well-justified, but it lacks details about the methodology followed and the implementation specifics.

Experimental design

The experimental design is not well explained. In more details:
- What prompt did you use for the GPT models? Was there a preliminary prompt engineering step? Is in-context learning and few-shot learning being used? Do you use approaches like chain-of-thought?

- Why only GPT family LLMs? It would be interesting to see the performance of other similar prompt-based LLMs, e.g., Gemini1.5Flash, Mistral8B, etc.

- Why were no domain-specific language models for surgery or medicine used, given that the task is surgery-related? A comparison between general-English LLMs like GPT and smaller but domain-specific ones, such as "ClinicalBert" and "SurgicBERTa," would significantly strengthen the paper. Discuss the reason for this choice in the paper and consider a quantitative comparison.

- It would also be interesting to compare the results with a naive method based on keywords present in the title or abstract to assess how challenging the task is.
- An error analysis would be interesting. When does the model fail? Why does it fail?

Validity of the findings

- It is unclear which GPT model was used. GPT3.5Turbo? GPT4? GPT4o? This should be explicitly described in the paper.
- I am perplexed that the inter-annotator agreement is so low. Why is that? Is the task particularly complex? What is the profile of the human reviewers? Are they domain experts? Do they have advanced English language skills?
- Given that the inter-annotator agreement is so low, I wonder why no rounds of reviewer discussions were conducted to jointly review the inclusion criteria and discuss cases of disagreement to establish a solid ground truth for comparison with GPT.

Additional comments

Some claims in the introduction are unsubstantiated. For example, the section: "There are also inherent risks [...] enhanced capabilities in natural language processing and understanding." Are there studies showing that GPT models are better than n-gram-based methods for this specific task?

---

## Round 0.2 · Major Revisions

As you can see, the authors still have substantial comments, and so this article needs further revisions.

Reviewer 1 ·

Basic reporting

Dear authors,
thank you for providing these revisions. They increased the quality of the paper substantially. However, some points of criticism remain.

1. Line 93 – 96: To my knowledge, models based on GPT also use tokens to capture patterns and relationships in text. Essentially, a GPT model can be considered an advanced form of an n-gram model. The primary distinction lies in the methodology: GPT employs a neural network to compute the probability of the next token, whereas traditional n-gram models rely on a count-based approach.
2. Line 96-98: Please provide a source for that claim.
3. Line 100-101: I think this sentence is missing a word.
4. Line 106-111: Are you talking about a specific GPT or GPT in general? Which models are more advanced than GPT?
5. Line 106; 112: You state “we sought to assess the ability of the GPT-model”. I think this statement is too general. You assess the capabilities of a specific GPT-model namely GPT4. Or which one is it?
6. Line 130: Please include a detailed outline of the study to guide the reader. Indicate what each section of the paper will cover.

Experimental design

7. Line 125-129: The research question is well-formulated but could be made more specific. Instead of using the term “a large language model,” I recommend explicitly referring to “the large language model GPT-4”.
8. Line 157-158; 166: Could you clarify why the script was fed either to OpenAI ChatGPT OR the GPT-4 API? Please elaborate on the methodology and decision-making process for this step.
9. Line 201-218: I am confused with the calculation of the metrics.
- First, you state: “Precision: 0.74” and “Positive predictive value (PPV): 0.74”. However, to my knowledge, precision and PPV are the same thing (what the metrics also show).
- You state a “Recall 0.89” of and a “Sensitivity for included papers of 0.89”. Again, to my knowledge, recall and sensitivity are the same metric.
- Lastly, you state, a “Negative predictive value (NPV): 0.84” meaning if the human reviewers decide to exclude a paper, there is an 84% change the GPT-model would exclude it as well.” However, you state “the sensitivity of excluded papers was 0.59, suggesting a moderate rate of correctly identifying non-relevant studies”. How does this go together?
Currently I have low faith in the reported metrics. This also due to the fact that it is not described in detail how every metric is calculated. Please revise the entire section and make clear what you calculated, what are the baselines, what do you mean exactly.
10. Line 213-216: What constitutes a combined human decision. In my mind I can create the following scenario:
- Human reviewer 1: NO
- Human reviewer 2: Yes
- Combined human decision as one reviewer: ?
- GPT4: Yes
- Outcome: ?
Please be clear with what you mean when making such statements.
11. Line 219-220: Now you introduce the metric “false positive”, wouldn’t that fit more to the text starting in line 201?
12. Line 224-225: Could you provide the narrower inclusion criteria referenced here? They do not appear to be included in the appendix. Please specify how they differ from the broader criteria mentioned earlier.

Validity of the findings

13. Line 256-264: You state that the justification provided by GPT4 was considered useful and that it provided rationale for the inclusion or exclusion decision based on each of your criteria. However, looking at Table 4 on of the major flaws of the current GPT models becomes very present. They simply cannot count.

Here some examples from Table 4:
- Y.Zhang; L. Peng; L. Thang 2023
GPT states ” the paper is a review article, which is explicitly excluded by criterion 3 of the exclusion criteria” --> Your exclusion criterion 3 is: “Studies focusing solely on pediatric populations.”
- Yoshida et al. 2023
GPT states ”although the abstract does not explicitly mention other complications such as aspiration pneumonia, delayed gastric emptying, malnutrition, dilation, mendelson syndrome, or prolonged length of stay or hospitalization, the focus on anastomotic leaks, which are a significant complication, suggests relevance to criterion 3 --> Your criteria is “Randomized controlled trials (RCTs), cohort studies, case-control studies, or prospective observational studies.
GPT states: “the study design is a propensity score-matching analysis, which is a type of observational study, thus fulfilling criterion 4” --> Your criterion 4 is “Articles published in English language.”
GPT states: “the article is published in english as per criterion 5” --> That is criterion 4, criterion 5 is “Studies with adult participants (age ≥18 years)”
GPT states: “and involves adult participants as indicated by the context of esophageal cancer surgery, satisfying criterion 6” --> There is no criterion 6!

This goes one similarly for the exclusion criteria and is also present in the row of Yang et al. 2023. The question is, how useful are those justifications? How correct are those justifications? At least this needs a very critical reflection on what the GPT is trying to do and in what sense it can make useful contributions.
14. Please include the human decision (reviewer 1, reviewer 2, joint decision) in Table 4.
15. Line 340-341: I am no expert in Canadian labor law but I would suspect that an employer does not only need to pay the minimum wage but also employer contributions, making the hourly rate even higher (https://www.hireborderless.com/post/how-much-does-it-cost-to-hire-an-employee-in-canada).
16. Now that a research question has been explicitly formulated, the paper must also provide a clear and direct answer to it in the discussion or conclusion.

Additional comments

I think that your paper has potential. However, I still have concerns regarding the accuracy of the metrics and the clarity of the methodology in general. Those concerns must be addressed to ensure credibility and reproducibility of the study.

Reviewer 3 ·

Basic reporting

REVIEWER 3 - COMMENT #2
Since the prompt is essential to the paper, it should be directly included and discussed within the text, rather than relying solely on external citation. Please add and discuss it.

REVIEWER 3 - COMMENT #4
In the marked manuscript, lines 400 and 401 that you referenced appear to be missing, as the numbering jumps from 396 to 410. This gap prevents me from fully addressing your comment. You may be referring instead to lines 427-430 in the marked manuscript (though it's possible that line numbers vary depending on the software, as your file is in .dox format rather than .pdf). However, I would have expected a more in-depth and "concrete" discussion.

REVIEWER 3 - COMMENT #5
The task I suggested was precisely to evaluate the complexity of the task: if a naive model based on keywords can achieve similar performance, it probably makes no sense to use a model with billions of parameters to do the same thing. What are the most recurring words in the abstracts or titles that passed the inclusion tests? Is there any recurring pattern?

REVIEWER 3 - COMMENT #6
Also here, I would have expected a more in-depth, "concrete" and careful discussion.

REVIEWER 3 - COMMENT #7
The answer is unrelated to the question. It seems more like an answer to the previous question.

Overall the paper is improved, but the above points are still open.
Also, there are typos in the manuscript: I recommend a careful review of these aspects.

Experimental design

Add the prompt the Python script uses to perform the task.

Validity of the findings

no comment

---

## Round 0.3 · Minor Revisions

Dear Authors,

Notwithstanding the fact that the revised manuscript has undergone a process of refinement, it remains the case that it does not fully address the suggestions, changes, additions, modifications and comments of the two referees. It is expected that the manuscript will be resubmitted with minor revisions as per the recommendations of Reviewer 1, with particular attention to the points identified by Reviewer 3 as being of particular importance.

Best wishes,

Reviewer 1 ·

Basic reporting

Dear authors,
thank you for providing this second revision, It increased the quality of the paper even more. Still, some minor points of criticism remain.

1. You state:
“In fact, recently, Guo et al. proposed leveraging the OpenAI The study demonstrated that using GPT-4 for title and abstract screening yielded promising results, achieving a macro F1 score of 0.60 and a sensitivity of 0.76 for included papers (12).”
--> There appears to be a missing word or phrase after "leveraging the OpenAI." Please clarify or complete this sentence to maintain clarity.

2. You state:
“This approach enabled us to rigorously evaluate the LLM’s GPT-4 models effectiveness under varied levels of stringency, providing insights into its adaptability and robustness in more challenging conditions.”
--> Could you confirm if this sentence is grammatically, correct?

3. You state:
“Animal studies are excluded from the search results”
--> You mention this twice in this chapter, consider removing the duplicate mention.

4. You state:
- “Similar to the perioperative iteration, the, 1967 results from the literature search underwent title and abstracts were analyzed by 2 independent screeners and the GPT-model”
“The sensitivity for excluded papers it was 0.16”
--> Please carefully review the entire manuscript for typos and grammatical errors.

Experimental design

5. You state:
“Compared to other LLMs, the GPT offers is an insightful model to evaluate due to its status as one of the most widely used and recognized LLMs, offering both powerful natural language processing capabilities and user-friendly accessibility (6). Despite the availability of more advanced models, GPT’s broad acceptance and convenience for users were key factors in its selection for examining systematic review processes in this study.”
--> The phrase "the GPT" is vague. Are you specifically referring to OpenAI’s ChatGPT or GPT models in general? Consider clarifying to avoid confusion.

6. You state:
“For this investigation solely the title and abstract suitability was inputted to OpenAI ChatGPT or GPT-4 API to determine suitability”
--> It is still unclear to me why title and abstract where inputted either into OpenAI ChatGPT OR the GPT-4 API. Didn’t you use a script for testing the studies? So it was all API then?

7. You state:
- “All title and abstracts were manually screened by 3 independent reviewers (R.R, J.R, E.R)”
- “For the purposes of evaluating agreement, a "combined human decision" was defined as the consensus decision reached by the two independent human reviewers after resolving any discrepancies through discussion”
- “Similar to the perioperative iteration, the, 1967 <Missing word> results from the literature search underwent title and abstracts were analyzed by 2 independent screeners and the GPT-model”
--> Please clarify whether three or two reviewers were involved in these processes.

Validity of the findings

8. You state:
- “The sensitivity for excluded papers it was 0.16 …”
- “The sensitivity of excluded papers was 0.59 for the perioperative run and 0.16 for the preoperative run, indicating limited promise in excluding irrelevant papers.”
- “Compared to Abstrackr, which had an overall sensitivity of 0.91, this study demonstrated an inclusion sensitivity of 0.89 for the perioperative run, indicating comparable performance.”
--> You mentioned sensitivity was removed and changed with recall, but, as shown, it still appears in the text. Please verify and revise as needed to avoid confusion.

Additional comments

9: When answering the research question, make sure to tell the reader that you are answering it. For example:
In regard to the proposed research question ("How effective is the GPT-4 model in identifying relevant studies during the title and abstract screening stage of systematic reviews, particularly when comparing broad (perioperative) versus narrow (preoperative") inclusion criteria for perioperative risk factors in esophagectomy?) it can be concluded that …

10. You state:
“The search strategy was designed and validated by a trained research librarian (RS) to identify studies that investigate perioperative risk factors for complications following esophagectomy (Appendix 1).”
--> I am not an expert in the Cochrane Library, but I find Appendix 1 quite confusing. Could it be clarified further to enhance its readability and coherence? Additionally, I would recommend that the editor assess whether this appendix provides sufficient insides into the search process.

Reviewer 3 ·

Basic reporting

1. The authors responded to the questions, but their answers were often careless, superficial, and chaotic. The revised paper still contains inconsistencies, omissions, and inaccuracies in the way the technical language is used and in the way results are presented and discussed.
2. It is still unclear what prompt was used and with which parameters. Table 4 provides an example of what was given as input and the expected output, but it still does not include the actual prompt or setup. The paper also states that the default parameters of OpenAI's APIs were used. These default parameters may change, and they should be explicitly declared for reproducibility. I fear the model and the APIs were treated as mere black boxes without investigating their functionality.
3. Unfortunately, there seems to have been an error in how my comments were presented to the authors in the previous revision. What I meant was that I would have expected a more detailed error analysis: When does the model fail? Why does it fail? Can you investigate this issue?
4. There is significant confusion in the way the tables are managed. For example, Table 1 is captioned as "Table 4" (see pages 36 and 37 of the .pdf file), and Table 3 is captioned as "Table 2" (pages 41 and 42 of the paper), etc. This makes the review process difficult.

Experimental design

no comment

Validity of the findings

no comment

---

## Round 0.4 · accepted · Accept

Dear Authors,

The invited reviewers have not responded to the invitation to review the revised manuscript. I have personally assessed the revision and can confirm that the paper has been sufficiently improved. I therefore believe that it is now ready for publication. However, before the production stage, I would be grateful if you would pay careful attention to the usage of blank characters, abbreviations, and referencing style in the paper.

Best wishes,